# MixGRPO: Unlocking Flow-based GRPO Efficiency with Mixed ODE-SDE

## Abstract

Although GRPO substantially enhances flow matching models in human preference alignment of image generation, methods such as DanceGRPO still exhibit inefficiency due to the necessity of sampling and optimizing over all denoising steps specified by the Markov Decision Process (MDP). In this paper, we propose **MixGRPO**, a novel framework that leverages the flexibility of mixed sampling strategies through the integration of stochastic differential equations (SDE) and ordinary differential equations (ODE). This streamlines the optimization process within the MDP to improve efficiency and boost performance. Specifically, Mix-GRPO introduces a sliding window mechanism, using SDE sampling and GRPO-guided optimization only within the window, while applying ODE sampling outside. This design confines sampling randomness to the time-steps within the window, thereby reducing the optimization overhead, and allowing for more focused gradient updates to accelerate convergence. Additionally, as time-steps beyond the sliding window are not involved in optimization, higher-order solvers are supported for sampling. So we present a faster variant, termed **MixGRPO-Flash**, which further improves training efficiency while achieving comparable performance. MixGRPO exhibits substantial gains across multiple dimensions of human preference alignment, outperforming DanceGRPO in both effectiveness and efficiency, with nearly 50% lower training time. Notably, MixGRPO-Flash further reduces training time by 71%.[1]

## 1 Introduction

Recent advances (Liu et al., 2025a; Xue et al., 2025; Liu et al., 2025b; Li et al., 2025; Xu et al., 2023) in Text-to-Image (T2I) tasks have demonstrated that probability flow models can achieve improved performance by incorporating Reinforcement Learning from Human Feedback (RLHF) (Ouyang et al., 2022) strategies during the post-training stage to maximize rewards. Specifically, methods (Liu et al., 2025a; Xue et al., 2025) based on Group Relative Policy Optimization (GRPO) (Shao et al., 2024), have recently been studied, achieving optimal alignment with human preferences.

Current GRPO methods in probability flow models, *e.g.,* Flow-GRPO (Liu et al., 2025a), Dance-GRPO (Xue et al., 2025), leverage Stochastic Differential Equations (SDE) sampling at every denoising step to introduce randomness, addressing reliance on stochastic exploration in RLHF. They model the entire denoising process as a Markov Decision Process (MDP) in a stochastic environment, using GRPO to optimize the state-action sequence. However, the need to optimize all denoising steps not only increases overhead but also leads to inconsistent gradient descent, resulting in inefficient training. Specifically, to compute the policy ratio, it is essential to perform full-step sampling independently with the old policy model $\pi_{\theta_{\text{old}}}$ and new one $\pi_\theta$. While DanceGRPO proposes to randomly select a subset of steps at a fixed ratio to optimize, our empirical analysis in Figure 1 demonstrates a substantial performance degradation as the subset size is reduced.

To address these issues, we propose MixGRPO, which achieves more efficient stochastic exploration while enabling optimization over fewer denoising steps. Specifically, we employ a mixed ODE-SDE strategy, applying SDE sampling to a denoising sub-interval and Ordinary Differential Equations (ODE) sampling to the rest, confining randomness to the SDE interval. In this way, fewer time-steps

---

[1]Code is available in the supplementary materials.

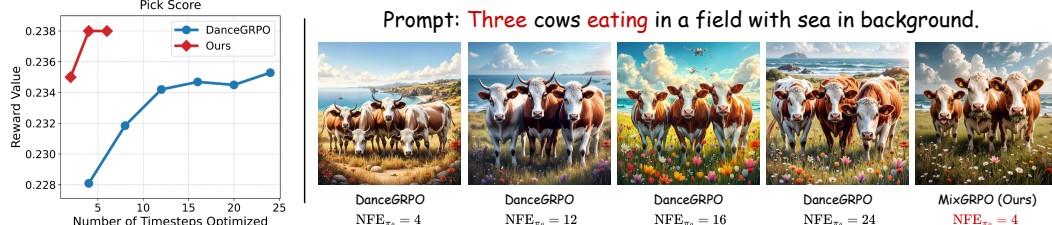

Figure 1: Performance comparison for different numbers of denoising steps optimized. The performance improvement of DanceGRPO relies on more steps optimized. MixGRPO achieves optimal performance while requiring only 4 steps.

are needed for GRPO optimization, without compromising image quality for reward computation. Besides, we introduce a sliding window strategy for the SDE interval that moves along the denoising steps as the training process progresses. Unlike random selection, this scheduling strategy orders optimization from high to low denoising levels, which aligns with the intuition of applying temporal discount factors to rewards in Reinforcement Learning (RL) (Pitis, 2019; Amit et al., 2020; Hu et al., 2022b). MixGRPO prioritizes optimizing the initial timesteps, which involve the most significant noise removal and entail a larger exploration space (see Figure 2). As training progresses, MixGRPO continuously moves the SDE window to narrow the exploration space and achieve global optimization. Finally, we find that higher-order ODE solvers, *e.g.,* DPMSolver++ (Lu et al., 2022b), can significantly accelerate training-time sampling, with negligible performance degradation, as there is no need for the posterior probability distribution after the sliding window.

We trained and evaluated MixGRPO by using HPS-v2.1 (Wu et al., 2023), Pick Score (Kirstain et al., 2023), ImageReward (Xu et al., 2023), and Unified Reward (Wang et al., 2025) as reward models (RMs) and metrics. We also quantified the overhead in terms of the number of function evaluations (NFE) and time consumption during training. During the training process, we fine-tuned based on FLUX.1-dev (Labs, 2024) and compared performance using either a single RM or multiple RMs as guidance, assessing the results for both in-domain and out-of-domain metrics. Specifically, trained and evaluated on the HPDv2 dataset (Wu et al., 2023), MixGRPO outperforms DanceGRPO across all metrics, particularly improving the ImageReward (Xu et al., 2023) from 1.088 to 1.629, surpassing DanceGRPO's score of 1.436, while generating images with enhanced semantic quality, aesthetics, and reduced distortion. Furthermore, MixGRPO reduces the training time of DanceGRPO by nearly 50%. In addition, MixGRPO-Flash utilizes DPMSolver++ (Lu et al., 2022b) to accelerate the sampling of $\pi_{\theta_{old}}$, reducing training time by 71%.

To summarize, the key contributions of our work are outlined below:

- We propose a mixed ODE-SDE GRPO training framework for flow-based models, which alleviates the overhead bottleneck by streamlining the optimization process within the MDP.

- We introduce a sliding window strategy to sample the denoising timesteps for model optimization, aligning with the RL intuition of transitioning from harder to easier search spaces, significantly enhancing performance.

- Our method enables the use of higher-order ODE solvers to accelerate $\pi_{\theta_{old}}$ sampling during GRPO training, achieving more significant speed improvements with comparable performance.

- Comprehensive experiments were carried out on multiple rewards and the results demonstrate that MixGRPO achieves substantial gains on various evaluation metrics, while significantly reducing training overhead.

## 2 RELATED WORK

### 2.1 RL FOR IMAGE GENERATION

Inspired by Proximal Policy Optimization (PPO) (Schulman et al., 2017), early works (Fan & Lee, 2023; Black et al., 2023; Fan et al., 2023; Lee et al., 2023) integrated reinforcement learning (RL) into diffusion models by optimizing the score function (Song et al., 2020b) through policy gradient

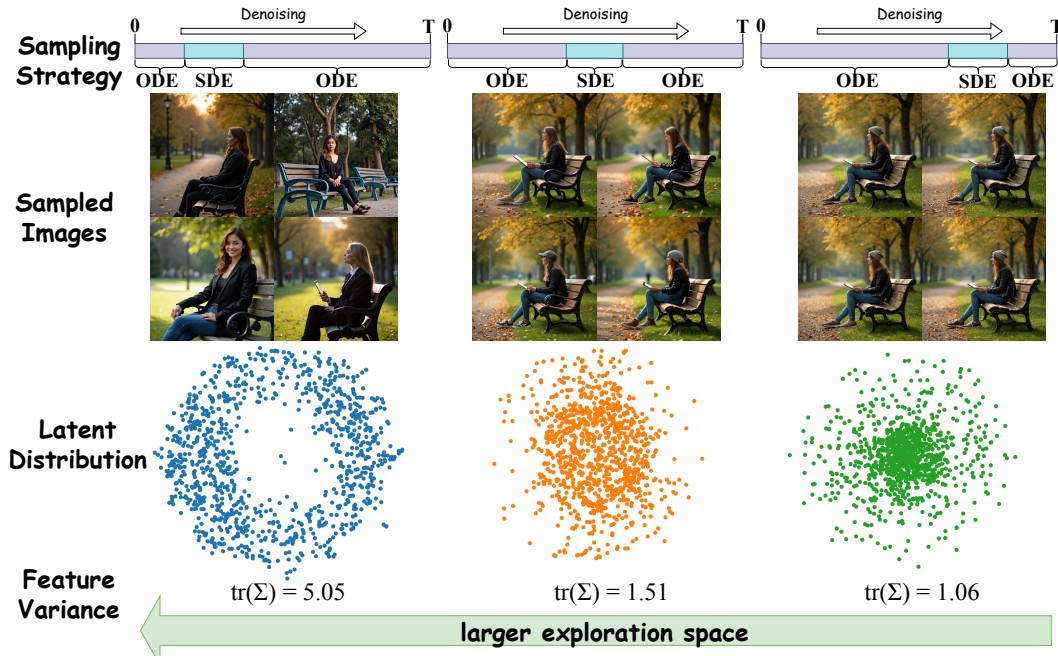

Figure 2: Visualization of t-SNE (Van der Maaten & Hinton, 2008) for images sampled with different strategies. Employing SDE sampling in the early stages of the denoising process results in a more discrete data distribution.

methods, thereby enabling the generation of images that better align with human preferences. Subsequently, Wallace et al. (2024) introduced offline-Direct Preference Optimization (DPO) to T2I tasks for the first time. This allows diffusion models to directly learn from human feedback and validates its effectiveness on large-scale models. Due to the tendency of offline win-lose pair data to shift the model away from its original distribution, some works (Yuan et al., 2024; Liang et al., 2025) have adopted online methods, continuously adjusting sampling trajectories through step-aware preference models during training to achieve improved performance. Recently, GRPO-based works *e.g.,* Tong et al. (2025), Flow-GRPO (Liu et al., 2025a) and DanceGRPO (Xue et al., 2025), have elevated RL-enhanced image generation to new heights. Specifically, Flow-GRPO (Liu et al., 2025a) and DanceGRPO (Xue et al., 2025) introduced GRPO to flow matching models, allowing divergent sampling by transforming the ODE into an equivalent SDE. They also identified the overhead caused by full-step sampling within a group as a bottleneck and sought to address it by reducing or randomly selecting denoising steps. However, these approaches do not fundamentally address the issue. We hope to delve into the essence of GRPO on the probability flow and provide deeper insights through mixed sampling techniques and optimization scheduling.

## 2.2 SAMPLING METHODS FOR PROBABILITY FLOW

DDPM (Ho et al., 2020) first proposed training a probabilistic model to reverse each step of noise corruption and utilized probability flow SDE for sampling, enabling the generation of realistic images. However, this often requires thousands of steps, resulting in significant overhead. DDIM (Song et al., 2020a) introduced deterministic sampling and proposed a probability ODE sampling approach, reducing the number of sampling steps to around 100. Subsequently, inspired by the Fokker-Planck equation (Risken & Risken, 1996), Song et al. (2020b) established a unification of SDE and ODE sampling methods from the perspective of the score function. Then, more higher-order ODE solvers were proposed *e.g.,* DPM-Solver (Lu et al., 2022a) and DPM-Solver++ (Lu et al., 2022b), which utilize multistep methods for differential discretization. These approaches significantly reduce the number of sampling steps to around 10 while preserving accuracy. Higher-performance solvers (Zheng et al., 2023; Zhao et al., 2023) continue to be proposed; however, the gains are relatively marginal and have ultimately been replaced by the distillation method (Salimans & Ho, 2022; Yin et al., 2024). During the same period, flow matching models (Lipman et al., 2022; Esser et al., 2024) simplified and stabilized training by predicting the vector field velocity, enabling deterministic

tic sampling with ODEs under 50 steps. Recent theoretical works (Gao et al., 2024; Albergo et al., 2023) has proven that the sampling method of flow matching is equivalent to DDIM, and demonstrated that flow matching models share the same equivalent SDE and ODE formulations as diffusion models. This provides important theoretical support and insights for our work, and we may explore interleaved sampling of SDE and ODE in probability flow models as a potential approach.

## 3 METHOD

### 3.1 MIXED ODE-SDE SAMPLING IN GRPO

According to Flow-GRPO (Liu et al., 2025a), the SDE sampling in flow matching can be framed as a Markov Decision Process (MDP) $(\mathcal{S}, \mathcal{A}, \rho_0, P, \mathcal{R})$ in a stochastic environment. The agent produces a trajectory during the discrete sampling process defined as $\Gamma = (\mathbf{s}_0, \mathbf{a}_0, \mathbf{s}_1, \mathbf{a}_1, \ldots, \mathbf{s}_T, \mathbf{a}_T)$, where the reward is provided only at the final step by the reward model, specifically $\mathcal{R}(\mathbf{s}_i, \mathbf{a}_i) \stackrel{\triangle}{=} R(\mathbf{x}_T, c)$ if $i = T$, and 0 otherwise.

In MixGRPO, we propose a hybrid sampling method that combines SDE and ODE. MixGRPO defines a time interval $S = [t_l, t_r) \in [0, 1)$, which corresponds to a subinterval of denoising timesteps, such that $0 \leq l < r \leq T$ and $t_i = \frac{i}{T}$. We use SDE sampling within the interval $S$ and ODE sampling outside, while $S$ shifts along the denoising direction throughout the training process (See Figure 2). MixGRPO restricts the agent's stochastic exploration space to the interval $S$, shortening the sequence length of the MDP to a subset $\Gamma_{\text{MixGRPO}} = (\mathbf{s}_l, \mathbf{a}_l, \mathbf{s}_{l+1}, \mathbf{a}_{l+1}, \ldots, \mathbf{s}_r, \mathbf{a}_r)$ and requires reinforcement learning (RL) optimization only on this subset:

$$\max_{\theta} \mathbb{E}_{\Gamma_{\text{MixGRPO}} \sim \pi_\theta} \left[ \sum_{t \in S} \left( \mathcal{R}(\mathbf{s}_t, \mathbf{a}_t) - \beta D_{KL} \left( \pi(\cdot|\mathbf{s}_t) \| \pi_{\text{ref}}(\cdot|\mathbf{s}_t) \right) \right) \right], \tag{1}$$

where $\mathcal{R}(\mathbf{s}_i, \mathbf{a}_i) \stackrel{\triangle}{=} R(\mathbf{x}_T, c)$ if $i = r$, and 0 otherwise. MixGRPO reduces computational overhead while also lowering the difficulty of optimization. Next, we derive the specific sampling form and optimization objective of MixGRPO.

For a deterministic reverse *probability flow ODE* (Song et al., 2020b), it takes the following form:

$$\frac{d\mathbf{x}_t}{dt} = f(\mathbf{x}_t, t) - \frac{1}{2}g^2(t)\nabla_{\mathbf{x}_t} \log q_t(\mathbf{x}_t), \quad \mathbf{x}_0 \sim q_0(\mathbf{x}_0), \tag{2}$$

where $q_t(\mathbf{x}_t)$ represents the evolution process of the reverse probability distribution from 0 to $T$. $\nabla_{\mathbf{x}_t} \log q_t(\mathbf{x}_t)$ is the *score function* at time $t$. According to the Fokker-Planck equation (Risken & Risken, 1996; Øksendal, 2003), Song et al. (2020b) has demonstrated that Eq. (2) has the following equivalent *probability flow SDE*, which maintains the same marginal distribution at each time $t$:

$$\frac{d\mathbf{x}_t}{dt} = f(\mathbf{x}_t, t) - g^2(t)\nabla_{\mathbf{x}_t} \log q_t(\mathbf{x}_t) + g(t)\frac{d\mathbf{w}}{dt}, \quad \mathbf{x}_0 \sim q_0(\mathbf{x}_0). \tag{3}$$

In MixGRPO, we mix ODE and SDE for sampling, which has the same convergence as using only ODE sampling (a detailed proof in Appendix A). The specific form is as follows:

$$d\mathbf{x}_t = \begin{cases} \left[ f(\mathbf{x}_t, t) - g^2(t)\nabla_{\mathbf{x}_t} \log q_t(\mathbf{x}_t) \right] dt + g(t)d\mathbf{w}, & \text{if } t \in S, \\ \left[ f(\mathbf{x}_t, t) - \frac{1}{2}g^2(t)\nabla_{\mathbf{x}_t} \log q_t(\mathbf{x}_t) \right] dt, & \text{otherwise.} \end{cases} \tag{4}$$

In particular, for Flow Matching (FM) (Lipman et al., 2022), especially the Rectified Flow (RF) (Liu et al., 2022), the sampling process can be viewed as a deterministic ODE:

$$\frac{d\mathbf{x}_t}{dt} = \mathbf{v}_t. \tag{5}$$

Eq. (5) is actually a special case of the Eq. (2) with $\mathbf{v}_t = f(\mathbf{x}_t, t) - \frac{1}{2}g^2(t)\nabla_{\mathbf{x}_t} \log q_t(\mathbf{x}_t)$. So we can derive the ODE-SDE hybrid sampling form for RF as follows:

$$d\mathbf{x}_t = \begin{cases} \left( \mathbf{v}_t - \frac{1}{2}g^2(t)\nabla_{\mathbf{x}_t} \log q_t(\mathbf{x}_t) \right) dt + g(t)d\mathbf{w}, & \text{if } t \in S, \\ \mathbf{v}_t dt, & \text{otherwise.} \end{cases} \tag{6}$$

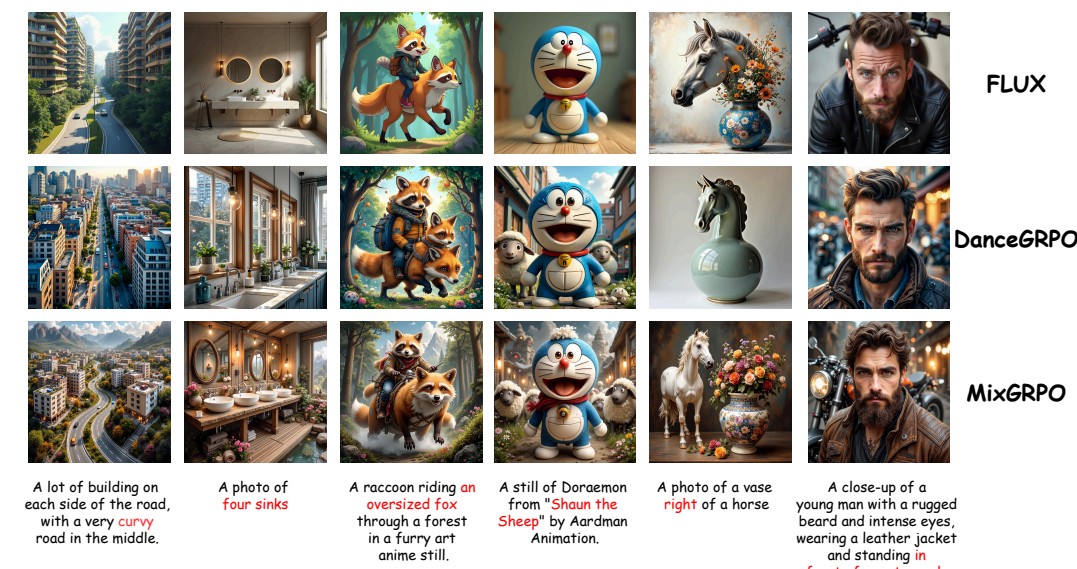

Figure 3: Qualitative comparison. MixGRPO achieve superior performance in semantics, aesthetics and text-image alignment.

In the RF framework, the model is used to predict the velocity field of the deterministic ODE, represented as $\mathbf{v}_\theta(\mathbf{x}_t, t) = \frac{d\mathbf{x}_t}{dt}$. Following Liu et al. (2025a), the *score function* is represented as $\nabla_{\mathbf{x}_t} \log q_t(\mathbf{x}_t) = -\frac{\mathbf{x}_t}{t} - \frac{1-t}{t}\mathbf{v}_\theta(\mathbf{x}_t, t)$. The $g(t)$ is represented as the standard deviation of the noise $g(t) = \sigma_t$. According to the definition of the standard Wiener process, we use $d\mathbf{w} = \sqrt{dt}\epsilon$, where $\epsilon \sim \mathcal{N}(0, \mathbf{I})$. Applying Euler-Maruyama discretization for SDE and Euler discretization for ODE, we build the final denoising process in MixGRPO:

$$\mathbf{x}_{t+\Delta t} = \begin{cases} \mathbf{x}_t + \left[\mathbf{v}_\theta(\mathbf{x}_t, t) + \frac{\sigma_t^2(\mathbf{x}_t + (1-t)\mathbf{v}_\theta(\mathbf{x}_t, t))}{2t}\right]\Delta t + \sigma_t\sqrt{\Delta t}\epsilon, & \text{if } t \in S \\ \mathbf{x}_t + \mathbf{v}_\theta(\mathbf{x}_t, t)\Delta t, & \text{otherwise} \end{cases} \tag{7}$$

According to Eq. (1), we only need to optimize GRPO (Shao et al., 2024) at the time within the interval $S$ for $N$ samples in the group. The final training objective is represented as follows:

$$\mathcal{J}_{\text{MixGRPO}}(\theta) = \mathbb{E}_{c \sim \mathcal{C}, \{\mathbf{x}_T^i\}_{i=0}^N \sim \pi_{\theta_{\text{old}}}(\cdot|c)}$$
$$\left[\frac{1}{N}\sum_{i=1}^N \frac{1}{|S|}\sum_{t \in S} \min\left(r_t^i(\theta)A^i, \text{clip}\left(r_t^i(\theta), 1-\varepsilon, 1+\varepsilon\right)A^i\right)\right], \tag{8}$$

where $r_t^i(\theta)$ is referred to as the policy ratio and $A^i$ is the advantage score. We set $\varepsilon = 0.0001$,

$$r_t^i(\theta) = \frac{q_\theta(\mathbf{x}_{t+\Delta t}|\mathbf{x}_t, c)}{q_{\theta_{\text{old}}}(\mathbf{x}_{t+\Delta t}|\mathbf{x}_t, c)} \quad \text{and} \quad A^i = \frac{R\left(\mathbf{x}_T^i, c\right) - \text{mean}\left(\{R\left(\mathbf{x}_T^i, c\right)\}_{i=1}^N\right)}{\text{std}\left(\{R\left(\mathbf{x}_T^i, c\right)\}_{i=1}^N\right)}, \tag{9}$$

It is important to note that, we have dropped the *KL Loss*. Although *KL Loss* can mitigate reward hacking to some extent (Liu et al., 2025a), inspired by yifan123 (2025), we use hybrid inference at test time, which can significantly address the reward hacking issue (See Appendix B).

MixGRPO reduces NFE of $\pi_\theta$ compared to all-timesteps optimization. However, the NFE of $\pi_{\theta_{\text{old}}}$ is not reduced, as complete inference is required to obtain the final image for reward calculation. In Section 3.3, we will introduce the use of higher-order ODE solvers, which also reduce the NFE of $\pi_{\theta_{\text{old}}}$ leading to further speedup. In summary, the mixed ODE-SDE sampling significantly reduces overhead while ensuring the introduction of randomness, allowing for optimization by GRPO.

### 3.2 SLIDING WINDOW AS OPTIMIZATION SCHEDULER

In fact, the interval $S$ can be non-fixed during the training process. In this section, we will introduce the sliding window to describe the movement of $S$, which leads to a significant improvement in

the quality of the generated images. Along the denosing time-steps $\{0, 1, \ldots, T-1\}$, MixGRPO defines a SDE sliding window $W(l)$ and optimization is only employed at the steps within $W(l)$.

$$W(l) = \{t_l, t_{l+1}, \ldots, t_{l+w-1}\}, \quad l \leq T - w, \tag{10}$$

where $l$ is the *left boundary* of the sliding window, and $w$ is a hyperparameter that represents the *window size*.

The *left boundary* $l$ of the sliding window moves as the training progresses. In the experiments, we found that the *window size* $w$, *shift interval* $\tau$, and *window stride* $s$ are all crucial hyperparameters. Through ablation studies (See Experiment 4.4.1), we identified the optimal settings. When the total sampling steps $T = 25$, the best performance is achieved with $w = 4$, $\tau = 25$ and $s = 1$. The detailed sliding window strategy and MixGRPO algorithm can be found in Algorithm 1.

Restricting the use of SDE sampling within the sliding window not only ensures the diversity of the generated images but also allows the model to concentrate on optimizing the flow within that window. Moving along the denoising direction represents the stochasticity of the probability flow from strong to weak, as illustrated in Figure 2. This is essentially a greedy strategy and is similar to the RL that assigns discount factors to process rewards (Pitis, 2019; Amit et al., 2020; Hu et al., 2022b), which gives greater importance to rewards derived from a larger search space in the earlier process.

The experimental results of different movement strategies in Table 4 demonstrate the validity of this intuition. Not only does the progressive strategy outperform random selection, but we also found that even when the sliding window is not moved (frozen), meaning only the earlier timesteps are optimized, MixGRPO can still yield good results, particularly in terms of ImageReward (Xu et al., 2023) and UnifiedReward (Wang et al., 2025). Based on this intuition, we also proposed an exponential decay strategy as follows, allowing $\tau$ to decrease as the window moves, enabling the model to avoid excessive optimization in smaller search spaces.

$$\tau(l) = \tau_0 \cdot \exp\left(-k \cdot \text{ReLU}\left(l - \lambda_{thr}\right)\right), \tag{11}$$

where $\tau_0$ is the initial shift interval, $k$ is the decay factor, and $\lambda_{thr}$ is the threshold that controls when the decay starts. The exponential function $\exp(x)$ calculates $e^x$, while the Rectified Linear Unit $\text{ReLU}(x)$ is defined as $\max(0, x)$. Table 4 shows that the exponential decay strategy can achieve better results in terms of Pick Score (Kirstain et al., 2023) and ImageReward (Xu et al., 2023). This may be because the model focuses on the earlier stages of denoising, which can lead to more significant high-level changes, precisely what the human preference alignment reward model emphasizes.

## 3.3 TRADE-OFF BETWEEN OVERHEAD AND PERFORMANCE

MixGRPO employs SDE sampling within the sliding window and ODE sampling outside of it, allowing the use of higher-order ODE solvers to accelerate GRPO training-time sampling. The timesteps that utilize ODE sampling are divided into those before and after the sliding window. The timesteps after the sliding window solely influence the reward calculation, whereas the timesteps before the window affect both the reward and contribute to cumulative errors in the policy ratio computing. Therefore, we focus exclusively on the acceleration of the timesteps after the window.

Gao et al. (2024) has demonstrated the equivalence between the ODE sampling of flow matching models (FM) and DDIM, and Section 3.1 has also shown that diffusion probabilistic models (DPM) and FM share the same ODE form during the denoising process. Therefore, the higher-order ODE solvers *e.g.,* DPM-Solver Series (Lu et al., 2022a;b; Zheng et al., 2023), UniPC (Zhao et al., 2023) designed for DPM sampling acceleration are also applicable to FM. We have reformulated DPM-Solver++ (Lu et al., 2022b) to apply it in the FM framework for ODE sampling acceleration and released detailed derivations in Appendix C.

By applying higher-order solvers, we achieve acceleration in the sampling of $\pi_{\theta_{\text{old}}}$ during GRPO training, which is essentially a balance between overhead and performance. Excessive acceleration leads to fewer timesteps, which inevitably results in a decline in image generation quality, thereby accumulating errors in the computation of rewards. We have found in practice that the 2nd-order DPM-Solver++ is sufficient to provide significant acceleration while ensuring that the generated images align well with human preferences in Table 8. Ultimately, we introduced DPM-Solver++

Table 1: Comparison results for overhead and performance. MixGRPO achieves the best performance across multiple metrics. MixGRPO-Flash significantly reduces training time while outperforming DanceGRPO. **Bold**: rank 1. Underline: rank 2. *The Frozen strategy means that optimization is only employed at the initial denoising steps.

| Method | $\text{NFE}_{\pi_{\theta_{old}}}$ | $\text{NFE}_{\pi_\theta}$ | Iteration Time (s)↓ | Human Preference Alignment | | | |
|---|---|---|---|---|---|---|---|
| | | | | HPS-v2.1↑ | Pick Score↑ | ImageReward↑ | Unified Reward↑ |
| FLUX | / | / | / | 0.313 | 0.227 | 1.088 | 3.370 |
| DanceGRPO | 25 | 14 | 291.284 | 0.356 | 0.233 | 1.436 | 3.397 |
| | 25 | 4 | 149.978 | 0.334 | 0.225 | 1.335 | 3.374 |
| | 25 | 4* | 150.059 | 0.333 | 0.229 | 1.235 | 3.325 |
| MixGRPO | 25 | 4 | 150.839 | **0.367** | **0.237** | **1.629** | **3.418** |
| MixGRPO-Flash | 16 (Avg) | 4 | 112.372 | 0.358 | 0.236 | 1.528 | 3.407 |
| | 8 | 4* | **83.278** | 0.357 | 0.232 | 1.624 | 3.402 |

and adopted both progressive and frozen sliding window strategies, proposing MixGRPO-Flash and MixGRPO-Flash*. A detailed description of the algorithm can be found in Appendix E. These approaches achieve a greater degree of acceleration compared to MixGRPO, while also outperforming DanceGRPO in terms of human preference alignment performance.

## 4 EXPERIMENTS

### 4.1 EXPERIMENT SETUP

**Dataset** We conduct experiments using the prompts provided by the HPDv2[2] dataset, which is the official dataset for the HPS-v2 benchmark (Wu et al., 2023). The training set contains 103,700 prompts; in fact, MixGRPO achieved good human preference alignment results after training one epoch with only 9,600 prompts. The test set consists of 400 prompts. The prompts are diverse, encompassing four styles: "Animation", "Concept Art", "Painting", and "Photo".

**Model** Following DanceGRPO (Xue et al., 2025), we use FLUX.1 Dev (Labs, 2024) as the base model, which is an advanced text-to-image model based on flow matching.

**Overhead Evaluation** For the evaluation of overhead, we use two metrics: the number of function evaluations (NFE) (Lu et al., 2022a) and the time consumption per iteration during training. The NFE is divided into $\text{NFE}_{\pi_{\theta_{old}}}$ and $\text{NFE}_{\pi_\theta}$. $\text{NFE}_{\pi_{\theta_{old}}}$ represents the number of forward propagation of the reference model for computing the policy ratio and generating images. $\text{NFE}_{\pi_\theta}$ is the number of forward propagation of the policy model solely for the policy ratio. Additionally, the average training time per GRPO iteration provides a more accurate reflection of the acceleration effect.

**Performance Evaluation** We used four multiple reward models in conjunction for GRPO, namely HPS-v2.1 (Wu et al., 2023), Pick Score (Kirstain et al., 2023), ImageReward (Xu et al., 2023) and Unified Reward Wang et al. (2025), both as reward guidance during training and as evaluation metrics. These metrics are all based on human preferences but emphasize different aspects. For example, ImageReward (Xu et al., 2023) highlights image-text alignment and fidelity, while Unified Reward (Wang et al., 2025) concentrates on semantics. DanceGRPO Xue et al. (2025) also demonstrates that using multiple reward models can achieve better results. To validate the robustness of MixGRPO, we also followed DanceGRPO and conducted additional comparisons using HPS-v2.1 as a single reward, and combining HPS-v2.1 (Wu et al., 2023) and CLIP Score (Radford et al., 2021) as multi-rewards.

### 4.2 IMPLEMENTATION DETAILS

For training-time sampling, we first perform a shift $\tilde{t} = \frac{t}{1-(\tilde{s}-1)t}$ on the uniformly distributed $t_i = \frac{i}{T}$ where $i = [0, \ldots, T-1]$, and then define $\sigma_t = \eta\sqrt{\frac{\tilde{t}}{1-\tilde{t}}}$. We set $\tilde{s} = 3$ and $\eta = 0.7$ as scale. We set $T = 25$ as the total sampling steps. For GRPO, the model generates 12 images for each prompt and clips the advantage to the range $[-5, 5]$. It is important to note that we use gradient accumulation over 3 steps, which means that during a single training iteration, there are

---
[2]https://huggingface.co/datasets/ymhao/HPDv2

$\frac{12}{3} = 4$ gradient updates. For the exponential decay strategy of the sliding window in Eq. ( 11), we empirically set $\tau_0 = 20$, $k = 0.1$, and $\lambda_{thr} = 13$. Furthermore, when multiple reward models are jointly trained with, each reward is assigned equal weight.

For training, all experiments are conducted on 32 Nvidia GPUs with a batch size of 1 and a maximum of 300 iterations. We use AdamW (Loshchilov & Hutter, 2017) as the optimizer with a learning rate of 1e-5 and a weight decay coefficient of 0.0001. Mixed precision training is used with the bfloat16 (bf16) format, while the master weights are maintained at full precision (fp32).

### 4.3 MAIN EXPERIMENTS

In the main experiment, the four human-preference-based rewards were aggregated according to advantages, as shown in Algorithm 1. We evaluated the overhead and performance of MixGRPO in comparison with DanceGRPO, with the results presented in Table 1. The official DanceGRPO uses $\text{NFE}_{\pi_\theta} = 14$; however, for fairness, we also tested DanceGRPO with $\text{NFE}_{\pi_\theta} = 4$. For MixGRPO-Flash, we evaluated both the progressive and frozen strategies, and to ensure fairness, we also applied the frozen strategy to DanceGRPO.

We selected multiple scene prompts to visualize the results for FLUX.1 Dev, the officially configured DanceGRPO, and MixGRPO, as shown in Figure 3. It can be observed that MixGRPO achieved the best results in terms of semantics, aesthetics, and text-image alignment. Figure 5 shows the comparison results of DanceGRPO, MixGRPO, and MixGRPO-Flash with $\text{NFE}_{\pi_\theta} = 4$. It can be observed that under the same overhead conditions, MixGRPO achieved better results compared to DanceGRPO. Additionally, MixGRPO-Flash enables accelerated sampling of $\pi_{\theta_{\text{old}}}$, and as the overhead decreases, the quality of the generated images still maintains a strong alignment with human preferences.

Following DanceGRPO (Xue et al., 2025), we also trained and evaluated the model using a single reward model, *e.g.,* HPSv2.1 and two reward models, *e.g.,* HPSv2.1 and CLIP Score on the HPDv2 dataset (Wu et al., 2023). The results (See Table 2) demonstrate that MixGRPO achieves the best performance on both in-domain and out-of-domain rewards, whether using a single or multiple reward models. The visualized results are displayed in Figure 6 and Figure 7 of Appendix F.

To demonstrate the robustness of the MixGRPO, we followed Flow-GRPO (Liu et al., 2025a) and applied the Low-Rank Adaptation (LoRA) (Hu et al., 2022a) method on the Stable Diffusion 3.5 (SD3.5) (Esser et al., 2024). We used HPS-v2.1 (Wu et al., 2023), Pick Score (Kirstain et al., 2023), and ImageReward (Xu et al., 2023) as multi-rewards, comparing Flow-DPO (offline/online) (Liu et al., 2025b), and Flow-GRPO (Liu et al., 2025a). The results in Table 3 showed that MixGRPO is well-suited for the LoRA, significantly reducing training overhead while outperforming both Flow-DPO and Flow-GRPO in terms of speed and performance. The visualization results can be found in Figure 8 of Appendix F.

Table 2: Comparison results demonstrate that MixGRPO achieves the best performance on both in-domain and out-of-domain rewards.

| Reward Model | Method | In Domain | | Out-of-Domain | | |
|---|---|---|---|---|---|---|
| | | HPS-v2.1 | CLIP Score | Pick Score | ImageReward | Unified Reward |
| / | FLUX | 0.313 | 0.388 | 0.227 | 1.088 | 3.370 |
| HPS-v2.1 | DanceGRPO | 0.367 | 0.349 | 0.227 | 1.141 | 3.270 |
| | MixGRPO | **0.373** | **0.372** | **0.228** | **1.396** | **3.370** |
| HPS-v2.1 & CLIP Score | DanceGRPO | 0.346 | 0.400 | 0.228 | 1.314 | 3.377 |
| | MixGRPO | **0.349** | **0.415** | **0.229** | **1.416** | **3.430** |

Table 3: Comparison results demonstrate that MixGRPO outperforms Flow-DPO and Flow-GRPO in training efficiency and performance.

| Model | RL Method | $\text{NFE}_{\pi_{\theta_{old}}}$ | $\text{NFE}_{\pi_\theta}$ | HPSv2.1 | Pick Score | ImageReward |
|---|---|---|---|---|---|---|
| SD3.5-M | / | / | / | 0.3066 | 0.2266 | 1.1630 |
| SD3.5-M+DPO LoRA | Offline DPO | 40 | 40 | 0.3043 | 0.2220 | 1.4524 |
| | Online DPO | 40 | 40 | 0.3132 | 0.2210 | **1.5001** |
| SD3.5-M+GRPO LoRA | Flow-GRPO | **10** | 10 | 0.3312 | 0.2318 | 1.4572 |
| | MixGRPO | **10** | **4** | **0.3416** | **0.2360** | 1.4854 |

### 4.4 ABLATION EXPERIMENTS

#### 4.4.1 SLIDING WINDOW HYPERPARAMTERS

As introduced in Section 3.2, the *moving strategy*, *shift interval* $\tau$, *window size* $w$ and *window stride* $s$ are all important parameters of the sliding window. We conducted ablation experiments on each of them. For the moving strategy, we compared three approaches: *frozen*, where the window remains stationary; *random*, where a random window position is selected at each iteration; and *progressive*, where the sliding window moves incrementally with the denoising steps. For the *progressive* strategy, we tested different scheduling strategies where the interval $\tau$ initially starts at 25 but changes

with training iterations. As shown in Table 4, the results indicate that under the *progressive* strategy, either exponential decay or constant scheduling strategies are optimal. For the *shift interval* $\tau$, 25 is the optimal setting (See Table 5).

The number of inferences for $\pi_\theta$ increases with the growth of the window size $w$, leading to greater time overhead. We compared different settings of $w$, and the results are shown in Table 6. Ultimately, we selected $w = 4$ as a balanced setting between overhead and performance. For the *window stride* $s$, we found through experimentation that $s = 1$ is the optimal choice, as shown in Table 7.

Table 4: Comparison for moving strategies.

| Strategy | Interval Schedule | HPS-v2.1 | Pick Score | ImageReward | Unified Reward |
|---|---|---|---|---|---|
| Frozen | / | 0.354 | 0.234 | 1.580 | 3.403 |
| Random | Constant | 0.365 | 0.237 | 1.513 | 3.388 |
| Progressive | Decay (Linear) | 0.365 | 0.235 | 1.566 | 3.382 |
| | Decay (Exp) | 0.360 | **0.239** | **1.632** | 3.416 |
| | Constant | **0.367** | 0.237 | 1.629 | **3.418** |

Table 5: Comparison for shift interval $\tau$.

| $\tau$ | HPS-v2.1 | Pick Score | ImageReward | Unified Reward |
|---|---|---|---|---|
| 15 | 0.366 | 0.237 | 1.509 | 3.403 |
| 20 | 0.366 | **0.238** | 1.610 | 3.411 |
| 25 | **0.367** | 0.237 | **1.629** | **3.418** |
| 30 | 0.350 | 0.229 | 1.589 | 3.385 |

Table 6: Comparison for window size $w$

| $w$ | NFE$_{\pi_\theta}$ | HPS-v2.1 | Pick Score | ImageReward | Unified Reward |
|---|---|---|---|---|---|
| 2 | 2 | 0.362 | 0.235 | 1.588 | **3.419** |
| 4 | 4 | 0.367 | 0.237 | **1.629** | 3.418 |
| 6 | 6 | **0.370** | **0.238** | 1.547 | 3.398 |

Table 7: Comparison for window stride $s$

| $s$ | HPS-v2.1 | Pick Score | ImageReward | Unified Reward |
|---|---|---|---|---|
| 1 | 0.367 | 0.237 | **1.629** | **3.418** |
| 2 | 0.357 | 0.236 | 1.575 | 3.391 |
| 3 | **0.370** | 0.236 | 1.578 | 3.404 |
| 4 | 0.368 | **0.238** | 1.575 | 3.407 |

### 4.4.2 HIGH ORDER ODE SOLVER

MixGRPO enables the possibility of accelerating ODE sampling with high-order ODE solvers by combining SDE and ODE sampling methods. We first conducted ablation experiments on the order of the solver, using DPM-Solver++ (Lu et al., 2022b) as the high-order solver with the *progressive* strategy. The results, as shown in Table 8, indicate that the second-order mid-point method is the optimal setting. Then, as described in Section 3.3, we compared two acceleration approaches. One is MixGRPO-Flash, which utilizes the *progressive* window moving strategy. The other is MixGRPO-Flash*, which employs the *frozen* moving strategy. They all achieve a balance between overhead and performance by reducing the number of ODE sampling steps after the sliding window. However, in practice, MixGRPO-Flash requires the window to move throughout the training process, which results in a shorter ODE portion being accelerated. Consequently, the acceleration effect of MixGRPO-Flash, on average, is not as pronounced as that of MixGRPO-Flash*.

Table 8: Comparison of performance across different-order solvers. The second-order Mid-point method achieves the best performance.

| Order | Solver Type | HPS-v2.1 | Pick Score | ImageReward | Unified Reward |
|---|---|---|---|---|---|
| 1 | / | **0.367** | 0.236 | 1.570 | 3.403 |
| 2 | Midpoint | 0.358 | **0.237** | **1.578** | **3.407** |
| | Heun | 0.362 | 0.233 | 1.488 | 3.399 |
| 3 | / | 0.359 | 0.234 | 1.512 | 3.387 |

Table 9: Comparison of different sampling steps for $\pi_{\theta_{\text{old}}}$. MixGRPO-Flash* achieves good performance even with few steps.

| Method | Sampling Overhead | | Human Preference Alignment | | | |
|---|---|---|---|---|---|---|
| | NFE$_{\pi_{\theta_{\text{old}}}}$ | Time per Image (s) | HPS-v2.1 | Pick Score | ImageReward | Unified Reward |
| DanceGRPO | 25 | 9.301 | 0.334 | 0.225 | 1.335 | 3.374 |
| MixGRPO-Flash | 19 (Avg) | 7.343 | 0.357 | 0.236 | 1.564 | 3.394 |
| | 16 (Avg) | 6.426 | **0.362** | **0.237** | 1.578 | 3.407 |
| | 13 (Avg) | 5.453 | 0.344 | 0.229 | 1.447 | 3.363 |
| MixGRPO-Flash* | 12 | 4.859 | 0.353 | 0.230 | 1.588 | 3.396 |
| | 10 | 4.214 | 0.359 | 0.234 | 1.548 | **3.430** |
| | 8 | **3.789** | 0.357 | 0.232 | **1.624** | 3.402 |

## 5 CONCLUSION

Although GRPO (Shao et al., 2024) has seen significant success in the language modality, it is still in the early stages of progress in vision (Tong et al., 2025; Xue et al., 2025; Liu et al., 2025a). Existing flow-based GRPO faces challenges such as low sampling efficiency and slow training. To address these issues, we proposed MixGRPO, a novel training framework that combines SDE and ODE sampling. This hybrid approach allows for focused optimization on the SDE sampling flow component, reducing complexity while ensuring accurate reward computation. Inspired by the decay factor in reinforcement learning (Hu et al., 2022b), we introduce a sliding window strategy for scheduling the optimized denoising steps. Experimental results confirm the effectiveness of our approach in both single-reward and multi-reward settings. Additionally, MixGRPO decouples the denoising stages for optimization and reward computation, enabling acceleration of the latter with high-order solvers. We further propose MixGRPO-Flash, which balances overhead and performance. We hope MixGRPO will inspire deeper research into post-training for image generation, contributing to the advancement of Artificial General Intelligence (AGI).

## 6 ETHICS STATEMENT

I read all respects with the ICLR Code of Ethics `https://iclr.cc/public/CodeOfEthics` and the research conducted in the paper complies in all respects.

## 7 REPRODUCIBILITY STATEMENT

This paper fully discloses all the source code needed to reproduce the main experimental results in the supplementary material. Besides, we also provide a detailed description of the MixGRPO algorithm (See algo. 1) and MixGRPO-Flash algorithm (See algo. 2). Finally, we also provide clear explanations of our proofs and derivations in sec. 3.1, app. A and app. C.

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

## A PROOF OF CONVERGENCE FOR MIXED ODE-SDE SAMPLING

To prove that the mixed ODE-SDE sampling method in Eq. (4) has the same convergence as Eq. (2), which uses only ODE sampling, referencing Song et al. (2020b), we approach this from the perspective of distribution evolution, where the distribution at each time step, *e.g.,* $\frac{\partial q_t(\mathbf{x})}{\partial t}$ must be the same. Let the interval for SDE be defined as $S = [t_l, t_r) \in [0, 1)$. Along the denoising direction, when the same initial Gaussian noise distribution $q_0(\mathbf{x}_0)$ is given, the probability distribution evolution in the ODE interval preceding the SDE is completely identical. The key point is whether the distribution evolution of the SDE within the interval $S$ is completely equivalent to that of the ODE. If they are equivalent, then the ODE interval following the SDE will naturally be equivalent to using only ODE sampling. Next, we will provide a detailed proof for this key point.

Consider the SDE Eq. (3) in the interval $S$, which possesses the following form:

$$\mathrm{d}\mathbf{x} = [f(\mathbf{x}, t) - g^2(t)\nabla_{\mathbf{x}} \log q_t(\mathbf{x})]\mathrm{d}t + g(t)\mathrm{d}\mathbf{w}, \quad t \in S. \tag{12}$$

The marginal probability density $q_t(\mathbf{x}_t)$ evolves according to Kolmogorov's equation (Fokker-Planck equation) (Øksendal, 2003)

$$\frac{\partial q_t(\mathbf{x})}{\partial t} = -\nabla_{\mathbf{x}} \cdot \left[ \left( f(\mathbf{x}, t) - g^2(t)\nabla_{\mathbf{x}} \log q_t(\mathbf{x}) \right) q_t(\mathbf{x}) \right] + \frac{1}{2}g^2(t)\nabla_{\mathbf{x}}^2 q_t(\mathbf{x}) \tag{13}$$

According to the definition of the Laplace operator $\nabla^2 h \equiv \nabla \cdot \nabla(h)$ and $\nabla_{\mathbf{x}} \log q_t(\mathbf{x}) = \frac{\nabla q_t(\mathbf{x})}{q_t(\mathbf{x})}$, we can obtain:

$$\begin{aligned}
\frac{\partial q_t(\mathbf{x})}{\partial t} &= -\nabla_{\mathbf{x}} \cdot \left[ f(\mathbf{x}, t)q_t(\mathbf{x}) - g^2(t)\nabla_{\mathbf{x}}q_t(\mathbf{x}) \right] + \frac{1}{2}g^2(t)\nabla_{\mathbf{x}}^2 q_t(\mathbf{x}) \\
&= -\nabla_{\mathbf{x}} \cdot [f(\mathbf{x}, t)q_t(\mathbf{x}) - \frac{1}{2}g^2(t)\nabla_{\mathbf{x}}q_t(\mathbf{x})] \\
&= -\nabla_{\mathbf{x}} \cdot \Big[ \underbrace{\left( f(\mathbf{x}, t) - \frac{1}{2}g^2(t)\nabla_{\mathbf{x}} \log q_t(\mathbf{x}) \right)}_{f_{\text{ODE}}(\mathbf{x}, t)} q_t(\mathbf{x}) \Big].
\end{aligned} \tag{14}$$

The Eq. (14) is indeed the Fokker-Planck equation of the ODE Eq. (2). Therefore, within the interval $S$, the distribution evolution of SDE and ODE sampling is consistent.

## B HYBRID INFERENCE FOR SOLVING REWARD HACKING

To address the reward hacking issue, we employ hybrid inference to sample with both the raw and post-trained models, and introduce the hybrid percent $p_{\text{mix}}$. This means that the initial $p_{\text{mix}}T$ denoising steps are sampled by the model trained with GRPO, while the remaining denoising process is finished by the original FLUX model (Black et al., 2023). Table 10 and Figure 4 respectively illustrate the changes in performance and images as $p_{\text{mix}}$ increases under the multi-rewards training scenario. The experimental results demonstrate that $p_{\text{mix}} = 80\%$ is an optimal empirical value that effectively mitigates hacking while maximizing alignment with human preferences.

Prompt: A painting depicting a snowy winter scene featuring a river, a small house on a hill, and a dreamy cloudy sky.

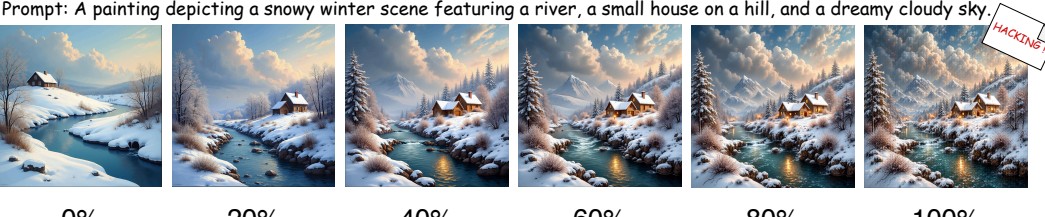

| 0% | 20% | 40% | 60% | 80% | 100% |

Figure 4: Qualitative comparison with different hybrid inference percentages

Table 10: Comparison with different hybrid inference percentages

| $p_{\text{mix}}$ | HPS-v2.1 | Pick Score | ImageReward | Unified Reward |
|---|---|---|---|---|
| 0% | 0.313 | 0.226 | 1.089 | 3.369 |
| 20% | 0.342 | 0.233 | 1.372 | 3.386 |
| 40% | 0.356 | 0.235 | 1.539 | 3.395 |
| 60% | 0.362 | 0.236 | 1.598 | 3.407 |
| 80% | 0.366 | **0.238** | **1.610** | **3.411** |
| 100% | **0.369** | 0.238 | 1.607 | 3.378 |

## C DPM-SOLVER++ FOR RECITIFIED FLOW

For clarity and to avoid ambiguity between continuous time and discrete steps, we adopt the following notation in this section. We denote the discrete time steps by an index $i \in \{0, 1, \ldots, T-1\}$, where $T$ is the total number of sampling steps. The continuous time corresponding to step $i$ is denoted by $t_i = \frac{i}{T} \in [0, 1)$.

The DPM-Solver++ algorithm (Lu et al., 2022b) is originally designed for the $\mathbf{x}_0$-*prediction* diffusion model (Rombach et al., 2022), where the model outputs the denoised feature $\mathbf{x}_0$ based on the noisy feature $\mathbf{x}_{t_i}$, the time condition $t_i$ and the text condition $c$. According to the definition of Rectified Flow (RF) (Liu et al., 2022), there is the following transfer equation:

$$\mathbf{x}_{t_i} = t_i \mathbf{x}_1 + (1 - t_i)\mathbf{x}_0. \tag{15}$$

According to the theory of stochastic interpolation (Albergo et al., 2023), RF effectively approximates $\mathbf{x}_1 - \mathbf{x}_0$ by modeling $\mathbf{v}_{t_i}$:

$$\mathbf{v}_{t_i} = \mathbf{x}_1 - \mathbf{x}_0. \tag{16}$$

Based on Eq. (15) and Eq. (16), we obtain the following relationship:

$$\mathbf{x}_0 = \mathbf{x}_{t_i} - \mathbf{v}_{t_i} t_i. \tag{17}$$

By using a neural network for approximation, we establish the relationship between RF and the $\mathbf{x}_0$-*prediction* model:

$$\mathbf{x}_\theta(\mathbf{x}_i, t_i, c) = \mathbf{x}_i - \mathbf{v}_\theta(\mathbf{x}_i, t_i, c) \cdot t_i. \tag{18}$$

Taking the multistep second-order DPMSolver++ as an example (see Algorithm 2 in (Lu et al., 2022b)), we derive the corrected $\mathbf{x}_\theta$ for the RF sampling process as $\mathbf{D}_i$:

$$\begin{aligned}\mathbf{D}_i \leftarrow &\left(1 + \frac{h_i}{2h_{i-1}}\right)\left(\mathbf{x}_{i-1} - \mathbf{v}_\theta(\mathbf{x}_{i-1}, t_{i-1}, c) \cdot t_{i-1}\right)\\ &- \frac{h_i}{2h_{i-1}}\left(\mathbf{x}_{i-2} - \mathbf{v}_\theta(\mathbf{x}_{i-2}, t_{i-2}, c) \cdot t_{i-2}\right),\end{aligned} \tag{19}$$

where $h_i = \lambda_{t_i} - \lambda_{t_{i-1}}$. The continuous time $t_i$ corresponds to the discrete step $i$ over a total of $T$ sampling steps. The term $\lambda_{t_i}$ is the log-*signal-to-noise-ratio* (log-SNR) and is defined in RF as:

$$\lambda_{t_i} := \log\left(\frac{1 - t_i}{t_i}\right). \tag{20}$$

Based on the exact discretization formula for the *probability flow ODE* proposed in DPM-Solver++ (Eq. (9) in (Lu et al., 2022b)), we can derive the final transfer equation:

$$\mathbf{x}_i \leftarrow \frac{t_i}{t_{i-1}}\mathbf{x}_{i-1} - (1 - t_i)\left(e^{-h_i} - 1\right)\mathbf{D}_i, \quad 1 \leq i < T. \tag{21}$$

## D  MIXGRPO ALGORITHM

Taking the progressive movement strategy of the sliding window as an example, the MixGRPO Algorithm 1 is as follows. MixGRPO first sets the left boundary $l$ of the sliding window $W(l)$ at $t = 0$ of the reverse process, which corresponds to the first sampling time-step. A GRPO iteration consists of two phases: sampling and optimization.

In the sampling phase, mixed ODE-SDE sampling is performed for the $N$ samples within the group, using SDE within the window and ODE outside. Then, the reward and advantage for the $N$ samples are calculated. For the multi-reward strategy, the advantage for each reward model is computed separately and then aggregated to obtain the final advantage. In the optimization phase, the GRPO loss (Shao et al., 2024) is computed and optimized only at the sampling timesteps within the SDE window $W(l)$.

After each GRPO iteration, the current training steps are checked against the conditions for moving the window, determining whether to shift the window by the window stride $s$.

---

**Algorithm 1** MixGRPO Training Process

---

**Require:** initial policy model $\pi_\theta$; reward models $\{R_k\}_{k=1}^K$; prompt dataset $\mathcal{C}$; total sampling steps $T$; number of samples per prompt $N$;
**Require:** sliding window $W(l)$, window size $w$, shift interval $\tau$, window stride $s$
1: Init left boundary of $W(l)$: $l \leftarrow 0$
2: **for** training iteration $m = 1$ **to** $M$ **do**
3:     Sample batch prompts $\mathcal{C}_b \sim \mathcal{C}$
4:     Update old policy model: $\pi_{\theta_{\text{old}}} \leftarrow \pi_\theta$
5:     **for** each prompt $\mathbf{c} \in \mathcal{C}_b$ **do**
6:         Init the same noise $\mathbf{x_0} \sim \mathcal{N}(0, \mathbf{I})$
7:         **for** generate $i$-th image from $i = 1$ **to** N **do**
8:             **for** sampling timestep $t = 0$ **to** $T - 1$ **do**                  ▷ $\pi_{\theta_{\text{old}}}$ mixed sampling loop
9:                 **if** $t \in W(l)$ **then**
10:                     Use SDE Sampling to get $\mathbf{x}_{t+1}^i$
11:                 **else**
12:                     Use ODE Sampling to get $\mathbf{x}_{t+1}^i$
13:                 **end if**
14:             **end for**
15:         **end for**
16:         **for** $i$-th image from $i = 1$ **to** N **do**
17:             Calculate multi-reward advantage: $A_i \leftarrow \sum_{k=1}^K \frac{R(\mathbf{x}_T^i, \mathbf{c})_k^i - \mu_k}{\sigma_k}$
18:         **end for**
19:         **for** optimization timestep $t \in W(l)$ **do**                  ▷ optimize policy model $\pi_\theta$
20:             Update policy model via gradient ascent: $\theta \leftarrow \theta + \eta \nabla_\theta \mathcal{J}$
21:         **end for**
22:     **end for**
23:     **if** $m \bmod \tau$ is 0 **then**                  ▷ move sliding window
24:         $l \leftarrow \min(l + s, \ T - w)$
25:     **end if**
26: **end for**

---

## E  MIXGRPO-FLASH ALGORITHM

MixGRPO-Flash Algorithm 2 accelerates the ODE sampling that does not contribute to the calculation of the policy ratio after the sliding window by using DPM-Solver++ in the Eq. (21). We introduce a compression rate $\tilde{r}$ such that the ODE sampling after the window only requires $(T - l - w)\tilde{r}$ time steps. And the total time-steps is $\tilde{T} = l + w + (T - l - w)\tilde{r}$ The final algorithm is as follows:

Note that when using MixGRPO-Flash*, the frozen strategy is applied, with the left boundary of the sliding window $l \equiv 0$. The theoretical speedup of the training-time sampling can be described as follows:

$$S = \frac{T}{w + (T - w)\tilde{r}}. \tag{22}$$

For MixGRPO-Flash, since the sliding window moves according to the progressive strategy during training, the average speedup can be expressed in the following form:

$$S = \frac{T}{\mathbb{E}_l \left( w + l + \lceil (T - w - l)\tilde{r} \rceil \right)} < \frac{T}{w + (T - w)\tilde{r}}. \tag{23}$$

---

**Algorithm 2** MixGRPO-Flash Training Process

---

**Require:** initial policy model $\pi_\theta$; reward models $\{R_k\}_{k=1}^K$; prompt dataset $\mathcal{C}$; total sampling steps $\tilde{T}$; number of samples per prompt $N$; ODE compression rate $\tilde{r}$
**Require:** sliding window $W(l)$, window size $w$, shift interval $\tau$, window stride $s$
1: Init left boundary of $W(l)$: $l \leftarrow 0$
2: **for** training iteration $m = 1$ **to** $M$ **do**
3:     Sample batch prompts $\mathcal{C}_b \sim \mathcal{C}$
4:     Update old policy model: $\pi_{\theta_{\text{old}}} \leftarrow \pi_\theta$
5:     **for** each prompt $\mathbf{c} \in \mathcal{C}_b$ **do**
6:         Init the same noise $\mathbf{x_0} \sim \mathcal{N}(0, \mathbf{I})$
7:         **for** generate $i$-th image from $i = 1$ **to** N **do**
8:             **for** sampling timestep $t = 0$ **to** $\tilde{T} - 1$ **do**
9:                 **if** $t < l$ **then**
10:                     Use first-order ODE sampling to get $\mathbf{x}_{t+1}^i$
11:                 **else if** $l \leq t < l + w$ **then**
12:                     Use SDE sampling to get $\mathbf{x}_{t+1}^i$
13:                 **else**                                             ▷ DPM-Solver++
14:                     Use higher-order ODE sampling to get $\mathbf{x}_{t+1}^i$
15:                 **end if**
16:             **end for**
17:         **end for**
18:         **for** $i$-th image from $i = 1$ **to** N **do**
19:             Calculate multi-reward advantage: $A_i \leftarrow \sum_{k=1}^K \frac{R(\mathbf{x}_{\tilde{T}}^i, \mathbf{c})_k^i - \mu_k}{\sigma_k}$
20:         **end for**
21:         **for** optimization timestep $t \in W(l)$ **do**                   ▷ optimize policy model $\pi_\theta$
22:             Update policy model via gradient ascent: $\theta \leftarrow \theta + \eta \nabla_\theta \mathcal{J}$
23:         **end for**
24:     **end for**
25:     **if** use MixGRPO-Flash* **then**                       ▷ move sliding window
26:         $l \leftarrow 0$
27:     **else**
28:         **if** $m \bmod \tau$ is 0 **then**
29:             $l \leftarrow \min(l + s, \ T - w)$
30:         **end if**
31:     **end if**
32: **end for**

---

# F   More visualized results

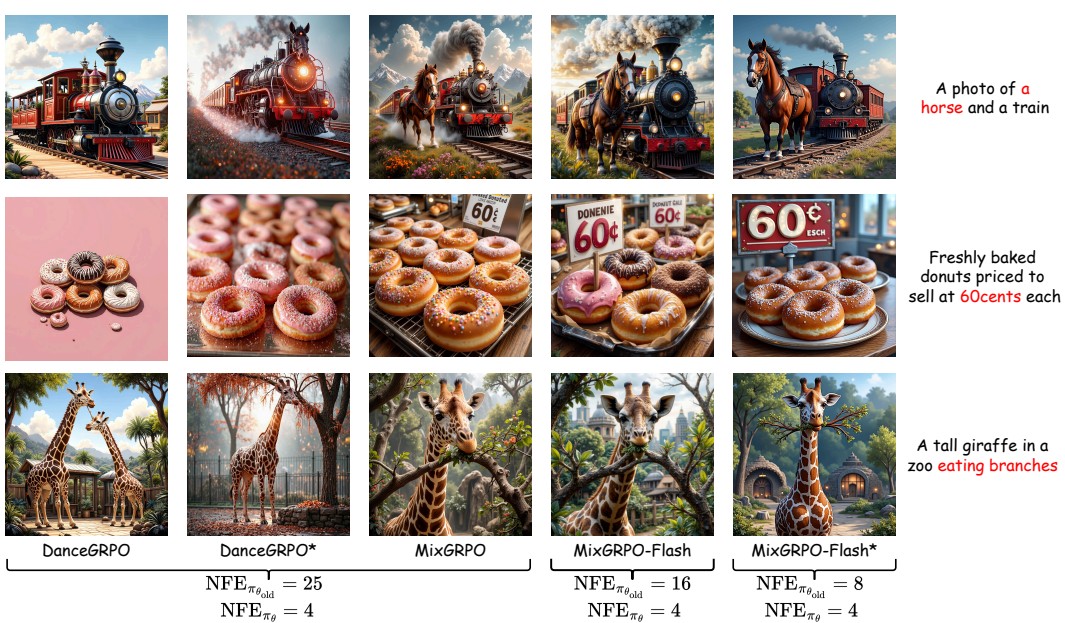

A photo of a horse and a train

Freshly baked donuts priced to sell at 60cents each

A tall giraffe in a zoo eating branches

| DanceGRPO | DanceGRPO* | MixGRPO | MixGRPO-Flash | MixGRPO-Flash* |

$$\text{NFE}_{\pi_{\theta_{\text{old}}}} = 25$$
$$\text{NFE}_{\pi_\theta} = 4$$

$$\text{NFE}_{\pi_{\theta_{\text{old}}}} = 16$$
$$\text{NFE}_{\pi_\theta} = 4$$

$$\text{NFE}_{\pi_{\theta_{\text{old}}}} = 8$$
$$\text{NFE}_{\pi_\theta} = 4$$

Figure 5: Qualitative comparison with different training-time sampling steps. The performance of MixGRPO does not significantly decrease with the reduction in overhead. *The Frozen strategy means that optimization is only employed at the initial denoising steps.

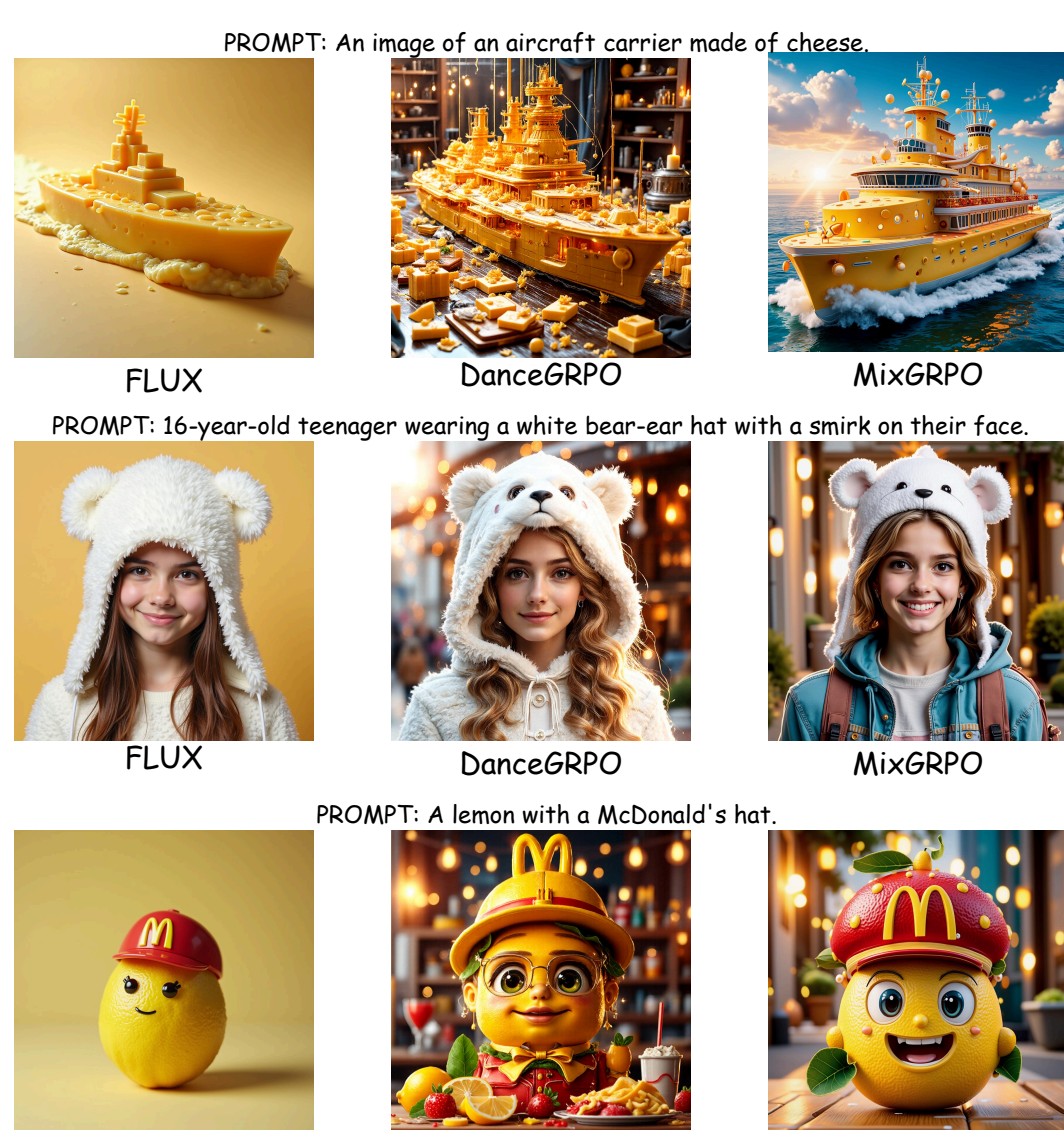

Figure 6: Comparison of the visualization results of FLUX, DanceGRPO, and MixGRPO under HPS-v2.1 as the reward model.

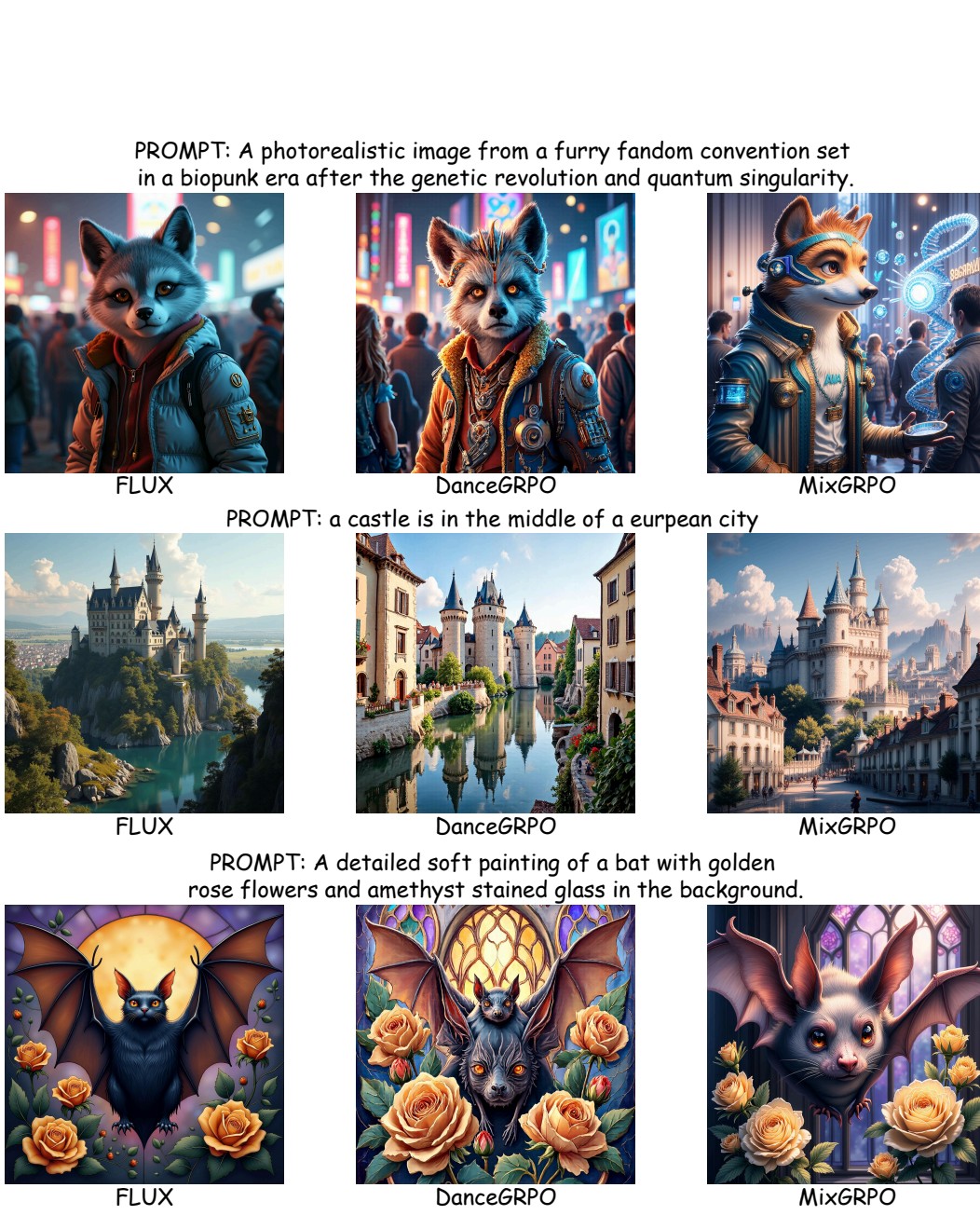

Figure 7: Comparison of the visualization results of FLUX, DanceGRPO, and MixGRPO under HPS-v2.1 and CLIP Score as multi-reward models.

PROMPT: a cute polar bear baby, digital oil painting by paul nicklen and by van gogh and monet

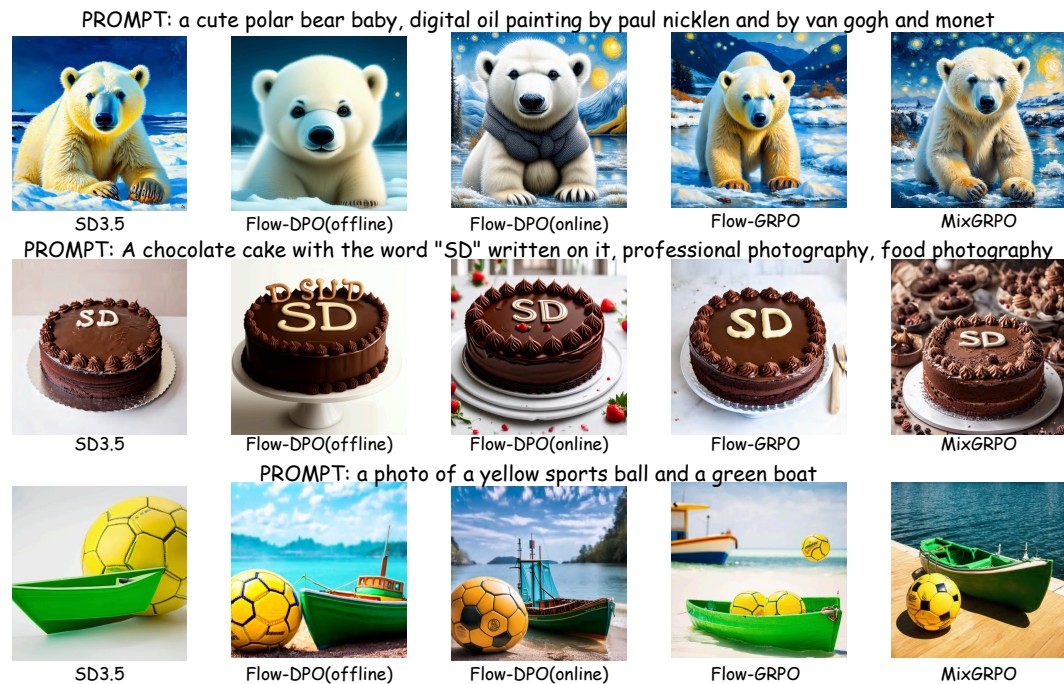

PROMPT: A chocolate cake with the word "SD" written on it, professional photography, food photography

PROMPT: a photo of a yellow sports ball and a green boat

Figure 8: Comparison of the visualization results of SD3.5-M, offline DPO, online DPO, Flow-GRPO and MixGRPO under HPS-v2.1, Pick Score and ImageReward as multi-reward models.

## G    THE USE OF LARGE LANGUAGE MODELS (LLMS)

LLMs were used solely to aid in writing and polishing the text (e.g., improving clarity and grammar), with all outputs verified by the authors.

