# OpenReview forum: "MixGRPO: Unlocking Flow-based GRPO Efficiency with Mixed ODE-SDE"
_ICLR.cc/2026/Conference — ICLR 2026 Conference Withdrawn Submission_

### Official Review · Reviewer_9TxA · 2025-10-29

**Soundness:** 3
**Presentation:** 2
**Contribution:** 2
**Rating:** 4
**Confidence:** 3

**Summary:**

The authors clearly articulate that existing flow‑based GRPO methods (e.g., DanceGRPO) sample stochastically at every denoising step, resulting in high overhead because both the old and the new policy must be sampled completely. They also observe that reducing the number of optimized steps in DanceGRPO leads to performance degradation. To tackle this, MixGRPO mixes SDE sampling on a sub‑interval of timesteps with ODE sampling elsewhere, thereby confining randomness to a sliding window and reducing the sequence length of the Markov Decision Process (MDP) for reinforcement learning; this is a principled way to limit exploration where it matters most.

**Strengths:**

- Technical design: The method is described formally. The authors derive a hybrid ODE–SDE sampling equation and show how it specializes to rectified flows, combining an SDE term within interval S and ODE outside. They then discretize the mixed dynamics using Euler‑Maruyama for the SDE part and Euler for the ODE part. A sliding window W(l) of fixed size w moves along the denoising steps to schedule which timesteps are optimized. Ablations examine different movement strategies (frozen, random, progressive) and window hyperparameters, showing that progressive schedules (constant or exponential decay) yield better human‑preference metrics.

- Efficiency improvements and results: By restricting SDE sampling to a small window and using ODE sampling elsewhere, MixGRPO reduces training time by nearly 50% relative to DanceGRPO while improving metrics such as HPS‑v2.1, Pick Score, ImageReward and Unified Reward. The variant MixGRPO‑Flash further uses high‑order ODE solvers (DPM‑Solver++) to speed up sampling of the old policy, reducing training time by ~71% while still outperforming baselines.

**Weaknesses:**

- Heuristic nature and hyperparameter sensitivity: MixGRPO relies heavily on several hyperparameters: the size of the SDE window w, the shift interval τ, the stride s, the movement strategy (frozen, random, progressive), and decay schedule parameters. Ablations show that performance varies substantially with these settings, indicating sensitivity. Yet the paper offers little guidance on analyzing robustness across datasets or models.

- Evaluation scope: Experiments focus on one flow‑matching base model (FLUX‑dev) and one LoRA version of Stable Diffusion 3.5, both fine‑tuned on the HPDv2 dataset. Metrics like ImageReward and Unified Reward are themselves learned models subject to bias; there is no human user study confirming that the improvements are perceptible. The qualitative results can be cherry-picked.

- The paper leverages hybrid inference to mitigate reward hacking, but this additional mechanism may confound the evaluation of MixGRPO itself. It would be important to disentangle the contribution of hybrid inference from that of MixGRPO to reveal the method’s true effect.

**Questions:**

- What is the main advantage of MixGRPO compared to recent reward-aligned distillation approaches such as [1] and [2]?
Additionally, could MixGRPO be applied to few-step distillation models?

- Have the authors conducted any human user-studies or human-rated benchmarks to validate that the improvements measured by automatic metrics correspond to perceptually meaningful gains?

- What are the results without hybrid inference? Does the baseline (DanceGRPO) in Table 1 also employ hybrid inference?
Ensuring identical inference settings is important for a fair comparison.

- Could the parameters—such as window size, shift interval, stride, or movement strategy—be adapted online during inference or training rather than tuned manually? A discussion on potential adaptive mechanisms or learning-based control would strengthen the practical contribution.

---
[1] ReNO: Enhancing One-step Text-to-Image Models through Reward-based Noise Optimization \
[2] Reward Fine-Tuning Two-Step Diffusion Models via Learning Differentiable Latent-Space Surrogate Reward

---

### Official Review · Reviewer_tS8j · 2025-10-30

**Soundness:** 2
**Presentation:** 3
**Contribution:** 2
**Rating:** 4
**Confidence:** 4

**Summary:**

This paper proposes MixGRPO, a framework that integrates SDE and ODE sampling within flow-matching models to improve the efficiency of GRPO-based reinforcement learning alignment for text-to-image generation. The method introduces a sliding-window mechanism that limits stochastic sampling (SDE) to a local interval while applying deterministic ODE sampling elsewhere. This design aims to reduce computation overhead while maintaining sufficient stochastic exploration. The paper also explores a progressive window schedule and a high-order ODE solver (DPM-Solver++) to further accelerate sampling. Experiments on the HPDv2 benchmark demonstrate superior reward-based metrics and reduced training time compared to Flow-GRPO and Dance-GRPO.

**Strengths:**

- The mixed sampling approach is an elegant idea that balances stochastic exploration and deterministic efficiency, representing a meaningful step beyond previous all-SDE GRPO variants.

- The progressive and decay-based movement of the SDE window is intuitively appealing, resembling a temporal discounting mechanism in RL, and empirically effective.

- Results show up to 50–70% faster training while achieving higher or comparable human-preference alignment scores.

- The authors perform ablation studies on hyperparameters ($w, \tau, s$) and test both progressive and frozen window strategies, showing consistent patterns.

**Weaknesses:**

- The proof in Appendix A only establishes marginal distribution equivalence between SDE and ODE sampling. It does not **guarantee unbiasedness or convergence** of the RL optimization when the window slides dynamically. Thus, the mixed process remains largely heuristic.
- The method introduces multiple coupled hyperparameters ($w, \tau , s$), and optimal values vary across metrics. This raises concerns about reproducibility and robustness across datasets or modalities.
- Although per-iteration training time is reduced, the required hyperparameter search and tuning may offset these gains in new tasks.

**Questions:**

- Does the marginal distribution equivalence in Appendix A imply that the policy-gradient estimates remain unbiased under mixed SDE–ODE sampling?
- When the window moves progressively, how does the algorithm prevent distributional mismatch between consecutive windows?
-  How sensitive are results to window size $w$ or shift interval  $\tau$ when transferring to a different dataset or total step count $T$?
- Author claims major efficiency gains—does this accounting include the potential hyperparameter search and validation time?

---

### Official Review · Reviewer_e3r4 · 2025-10-30

**Soundness:** 3
**Presentation:** 3
**Contribution:** 2
**Rating:** 4
**Confidence:** 2

**Summary:**

The paper proposes MixGRPO, an improved reinforcement learning method for flow-matching generative models. It introduces a sliding-window optimization scheme that mixes SDE and ODE sampling, effectively reducing training overhead while maintaining stochastic exploration. Also, a Flash version is incorporated to accelerate non-optimized steps, achieving substantial speedups with certain performance loss.

**Strengths:**

1. The paper presents a simple yet clear improvement on training speed over DanceGRPO by introducing a structured sliding-window mechanism and mixed ODE–SDE sampling.
2. By introducing high-order ODE solver, the authors propose a flash version of MixGRPO, which further accelerates sampling but compremising generation quality.

**Weaknesses:**

1. The authors remove the KL loss and rely on inference-time hybrid sampling to prevent reward hacking, but this design choice is not sufficiently analyzed. Ablation study would strengthen the claim of stability.
2. The evaluation inconsistently benchmarks against only one prior method, DanceGRPO in some tables, FlowGRPO in others, rather than both across all relevant experiments, hindering a comprehensive assessment of MixGRPO's advantages.

**Questions:**

Lines 142–144 claim that prior methods such as DanceGRPO suffer from high computational cost due to full-step sampling, and MixGRPO alleviates this by introducing a sliding-window mechanism. However, DanceGRPO itself already employs random substep optimization. Could the authors clarify what the essential difference is between the two approaches beyond the continuity of the sliding-window design? In particular, why does the proposed sliding window yield a substantial quality and efficiency gain if both methods ultimately optimize over subsets of timesteps?

---

> ### Author Response · Authors · 2025-11-13
> **Author Response**
>
> We sincerely thank you for taking the time to provide detailed and constructive feedback on our submission. Your comments are highly valuable and have significantly helped us identify areas for improvement in our work. Below, we address your suggestions and concerns.
>
> ### **Response:**
>
> **1. About removing the KL loss**
>
> Our optimization approach is developed based on DanceGRPO. It is noteworthy that the authors of DanceGRPO stated, "While traditional GRPO formulations employ KL-regularization to prevent reward over-optimization, we empirically observe minimal performance differences when omitting this component". Therefore, DanceGRPO removed KL Loss, and we observed the same phenomenon, so we retained this operation.
>
> **2. About inference-time hybrid sampling**
>
> In our study, we observed that both DanceGRPO and MixGRPO are prone to a reward hacking phenomenon during the later stages of training, which manifests as grid-like noise in the generated images as shown in Appendix B. Reward hacking is often an artifact of the reward model, which fails to provide a comprehensive assessment of image quality. As a result, optimizing for certain metrics can improve them at the expense of degrading performance in other areas. Therefore, this has nothing to do with the algorithm itself.
>
> To effectively mitigate this issue, we adopted the inference-time hybrid sampling method proposed in [1]. We have duly cited this work in the main text (Section 3.1) and provided a visual analysis in Appendix B.
>
> We wish to clarify that this inference-time hybrid sampling strategy is an effective technique we incorporated to enhance model robustness, rather than a novel contribution of our work. The core contributions of MixGRPO are centered on innovations within the training framework itself, which include:
> + A mixed ODE-SDE GRPO training framework.
> + A sliding window strategy to schedule the timesteps for GRPO optimization.
> + The use of higher-order ODE solvers to accelerate training-time rollout in GRPO.
> + Comprehensive experiments were carried out on multiple rewards and the results demonstrate that
> MixGRPO achieves substantial gains on various evaluation metrics, while significantly reducing
> training overhead.
>
> **3. About the comparison with DanceGRPO and FlowGRPO**
>
> Methodologically, DanceGRPO and FlowGRPO are largely analogous. Both methods convert the sampling process of a flow-matching model into an equivalent full-step SDE sampling, thereby introducing the necessary stochasticity for reinforcement learning exploration. While they differ slightly in their mathematical formulation, their primary distinction lies in the implementation details: DanceGRPO performs full fine-tuning on the Flux model, whereas FlowGRPO utilizes LoRA for fine-tuning on SD3.5-M.
>
> To ensure a fair comparison, we adopted the official configurations for each method. This approach allows us to demonstrate the robustness of MixGRPO, as it performs effectively across these distinct foundational models and fine-tuning strategies.
>
> **4. About the difference between "random substep optimization" (DanceGRPO) and "sliding-window design"(MixGRPO)**
>
> From a methodological standpoint, our sliding-window design offers a more structured approach to optimization. By shortening the effective length of the Markov Decision Process (MDP), it strategically confines the stochasticity of SDE sampling to a localized window. In contrast, the random substep optimization in DanceGRPO treats all denoised time steps as potential targets for the MDP. While this uniform random sampling effectively reduces computational overhead, it could not provide a focused signal for gradient updates, which can in turn hinder the rate of training convergence.
>
> Experimentally, this distinction becomes more apparent. We observe that random substep optimization's performance degrades significantly as step size decreases. Conversely, our sliding-window approach implicitly creates a training curriculum, analogous to reward shaping in RL. It prioritizes the optimization of high-noise regions, where the denoising process is inherently more stochastic, before refining areas with less noise, where the dynamics become more deterministic, as shown in Figure 2. This allows for more focused and efficient gradient updates, enabling the model to achieve superior performance with minimal overhead, as illustrated in Figure 1. To ensure a fair comparison under equivalent computational budgets, we evaluated both strategies with $\text{NFE}{\pi_{\theta_{\text{old}}}}=25$ and $\text{NFE}{\pi_{\theta}}=4$. As shown in Table 1, MixGRPO demonstrates a significant performance advantage over DanceGRPO under these controlled conditions.
>
> **I look forward to your response. Your timely response would be greatly appreciated !**
>
> [1] GitHub User yifan123. Discussion on flow-grpo issue 7. https://github.com/yifan123/flow_grpo/issues/7#issuecomment-2870678379, 2025.

---

### Official Review · Reviewer_onMn · 2025-11-01

**Soundness:** 3
**Presentation:** 3
**Contribution:** 3
**Rating:** 6
**Confidence:** 4

**Summary:**

This paper proposes MixGRPO, a hybrid reinforcement learning framework that combines SDE and ODE based sampling via a sliding-window mechanism to focus optimization on key timesteps, significantly improving efficiency and convergence while maintaining strong alignment with human preferences.

**Strengths:**

For reinforcement learning tasks, improving sampling efficiency is crucial. The authors propose a window-based sampling strategy that accelerates sampling while maintaining diversity.

**Weaknesses:**

In LLMs, it is rare to see algorithms that train using only partial tokens. Why is it acceptable in diffusion models to do so, and even achieve strong performance by optimizing only a frozen window? The paper lacks analysis and discussion on this aspect.

**Questions:**

1. The ablation study on the window scheduler reports results for a window size of 4. Would the conclusions change if more extreme window sizes (e.g., 1 or 2) were used?

2. For the same prompt, should the position of the SDE window remain fixed? If not, could varying the window position within a group of the same prompt enable the use of even smaller window sizes? I hope the authors can identify effective window configuration strategies that achieve strong performance even with a window size of 1.

3. Could the process be segmented, e.g., ODE–SDE–ODE–SDE–ODE, into multiple discontinuous windows? Would that yield better results?

4. In Figure 2, are the σ values used for visualization the same? If so, does this imply that, under the same injected noise level, higher-noise steps introduce greater randomness?

5. Why does introducing a higher-order solver in the ODE steps before the SDE stage affect the policy-ratio computation? As long as the inputs remain identical, the outputs should also be identical. In principle, higher-order solvers could be introduced before the SDE stage to accelerate computation.

If the authors can effectively address or answer my questions, I would be willing to give a higher score.

---

### Note · Authors · 2025-11-14

I have read and agree with the venue's withdrawal policy on behalf of myself and my co-authors.